# Learning from Adaptations to the COVID-19 Pandemic: How Teleconsultation Supported Cancer Care Pathways at a Comprehensive Cancer Center in Northern Italy

**DOI:** 10.3390/cancers15092486

**Published:** 2023-04-26

**Authors:** Giada Caviola, Jessica Daolio, Carlotta Pellegri, Francesca Cigarini, Luca Braglia, Marco Foracchia, Elisa Mazzini, Loredana Cerullo

**Affiliations:** 1Quality and Accreditation Office, Medical Directorate, Azienda Unità Sanitaria Locale-IRCCS di Reggio Emilia, 42123 Reggio Emilia, Italy; giada.caviola@ausl.re.it (G.C.); carlotta.pellegri@ausl.re.it (C.P.); francesca.cigarini@ausl.re.it (F.C.); loredana.cerullo@ausl.re.it (L.C.); 2Clinical Trials Center, Research and Statistics Infrastructure, Azienda Unità Sanitaria Locale-IRCCS di Reggio Emilia, 42123 Reggio Emilia, Italy; luca.braglia@ausl.re.it; 3Information Technology Unit Department, Azienda Unità Sanitaria Locale-IRCCS di Reggio Emilia, 42123 Reggio Emilia, Italy; marco.foracchia@ausl.re.it; 4Scientific Directorate, Azienda Unità Sanitaria Locale-IRCCS di Reggio Emilia, 42123 Reggio Emilia, Italy; elisa.mazzini@ausl.re.it

**Keywords:** cancer care pathway, COVID-19, multidisciplinary team meeting, oncology, teleconsultation

## Abstract

**Simple Summary:**

Multidisciplinary team (MDT) meetings are widely recognized as the gold standard for care management of cancer patients. During the pandemic, cancer care delivery was a priority to be maintained through cancer care pathways (CCPs) and MDT meetings, which were forcibly converted from in-person to telematic format. The aim of our retrospective study was to report the evolution of MDT meeting performance following the shift to teleconsultation by analyzing four MDT meeting indicators between 2019 and 2022. We observed that the MDT meeting teleformat strengthened the overall CCP performance by boosting the participation of MDT members and the number of discussed cases, all without compromising either the annual frequency or duration of MDT meetings. Considering the rapidity, extent, and intensity with which telematic tools have been adopted due to the COVID-19 pandemic, the results of this study help to understand the effects of these tools on health care and the parties involved.

**Abstract:**

Multidisciplinary team (MDT) meetings are recognized as the gold standard for care management of cancer patients, and during the COVID-19 pandemic they were considered a priority to be maintained. Due to pandemic-related restrictions, MDT meetings were forcibly converted from in-person to telematic format. This retrospective study evaluated the annual performance of four MDT meeting indicators (MDT members’ attendance, number of discussed cases, frequency of MDT meetings, and duration) between 2019 and 2022 to report on the implementation of teleconsultation in MDT meetings related to 10 cancer care pathways (CCPs). Over the study period, MDT member participation and the number of discussed cases improved or did not change in 90% (9/10) and 80% (8/10) of the CCPs, respectively. We did not observe significant differences in any of the CCPs included in the study regarding the annual frequency and duration of MDT meeting. Considering the rapidity, extent, and intensity with which telematic tools were adopted due to the COVID-19 pandemic, the results of this study showed that MDT teleconsultation supported the CCPs, and consequently, the delivery of cancer care in COVID-19 times, helping to understand the effects of telematic tools on health care performance and the parties involved.

## 1. Introduction

Care pathways (CPs) are complex interventions aimed at addressing discrepancies in the quality of care across the continuum [1]. A CP is concerned with a timeline of structured events to assist health care providers in coordinating and delivering care in the safest, most appropriate, effective, timely, efficient and equal manner [2,3,4]. It is also a valuable and cost-effective tool, especially in the field of oncology [5,6]. This characteristic is relevant in COVID-19 times, in which more professional and technical resources than ever need to be preserved. 

Meetings among multidisciplinary team (MDT) members represent the cornerstone of CPs, independently of the disease of interest. An MDT is composed of health care practitioners with specialized skills who meet periodically to carry out a decision-making process about diagnosis and treatment in accordance with evidence-based recommendations [7]. In newly diagnosed cancer patients, MDT members are involved at the outset of care to sub-classify the disease process and determine the intensity and type of therapy in order to avoid time-consuming and inadequate treatments and to ensure proper evidence-based management, while facing a complex landscape of treatment options [8]. In patients affected by colorectal, lung, prostate, and breast cancer, a recent review by Kočo et al. [9] showed that a case management plan can change in up to 58% of cases after MTD discussion, with a reduction in overall performed surgery and an increase in the use of chemotherapy and MRI imaging [9]. The authors also underscored that MDT discussion contributes to increasing the survival rate significantly, but provided weak evidence [9,10]. Weakness regarding this issue is also reported in head and neck cancer [11,12,13], gastroesophageal and gastrointestinal cancer [14,15,16,17], primary and/or recurrent vulvar carcinoma [18,19], bladder cancer [20], adrenocortical carcinoma [21], and ovarian cancer [22]. 

Based on the key role of MTD meetings in affecting patient outcomes, meetings discussing newly diagnosed cancer cases and/or cases of recurrence are widely recognized as the gold standard in CCP management and as a platform to achieve clinical integration [17,23,24]. However, factors still exist that influence the quality and functioning of MDT meetings with regard to members’ compliance and attendance, discrepancies between workload and health care professional resources, equipment availability, meeting format, communication practices, and the lack of awareness regarding the educational functions for residents [25,26,27]. To date, solutions able to overcome these barriers are still lacking, and such solutions are required in order to reach a standardized model by which an MDT meeting should be conducted [28]. In regard to the period of COVID-19, it is reasonable to speculate that the reorganization of health care services and strategies implemented in response to the major waves of the pandemic may have strengthened MDT barriers and consequently affected the functioning of CPs. In particular, health care providers quickly adopted telematic resources to provide and support health care when the pandemic forcibly separated the parties involved in the process of care. Teleconferencing has experienced unexpected implementation for sharing knowledge with the scientific community [29], and telemedicine, defined as a medical act in which the health care practitioner interacts remotely in real time with the patient [30], promptly supported many medical settings without compromising either medical care or the patient-clinician relationship [31,32,33,34,35]. To date, the effects of implementation of telematic resources are still under research, mainly because of the rapidity, extent, and intensity with which they were adopted. 

Our context is a public Local Health Authority (AUSL) comprising six hospitals together with a Cancer Research Hospital (IRCCS) that is officially recognized as a research hospital by the Italian Ministry of Health and designated as a comprehensive cancer center by the Organization of European Cancer Institutes (OECI). Both the province and region where our reality is located were hit severely and early by the first wave of the pandemic. To face the challenges the pandemic posed, our health system implemented a specific strategy based on the following key elements: building separate pathways for COVID and non-COVID patients; treating all COVID patients in need of hospitalization, while continuing to ensure adequate and appropriate care for all non-COVID patients in case of emergency and/or complex cases; never stop delivery of cancer care; and moving promptly and in a coordinated manner within the network and in cooperation with other local health organizations. As reported by researchers at our center [36], screening programs were suspended from the middle of March to the end of May, on average. However, once resumed, actions implemented to screen patients scheduled in that period minimized the impact of the lockdown on cancer screening delays and diagnosis. New diagnoses returned to a number only slightly lower than those observed in 2019. 

During the pandemic, the delivery of cancer care by means of eleven cancer CPs (CCPs) active in our context was a priority to be maintained over other services, including the continuation of MDT meetings. The methodological rules for MDT operation adopted by our AUSL-IRCCS are presented in Appendix A. Based on the pivotal role of MDT meetings for cancer care management, the Medical Directorate determined that CCP MDT members would be among the first to be provided with the limited teledevices available at the time. In parallel, CCP MDT members asked to be enabled to continue their meetings, affirming their indispensable value even in a situation of extreme rationing of resources. The devices for teleconferencing were actually already available at the Institution in the years prior to the pandemic, but only the prostate CCP MDT members used them. For this reason, the prostate CCP was not included in this study. The Quality and Accreditation Office supports CP activities according to the methodological model of the European Pathway Association [1] and through professional data managers, who collect data and information useful for monitoring the performance of CPs as daily practice and support professionals through the implementation. During the pandemic, CCP data managers maintained the daily practice and trained MDT groups on the installation and use of the digital discussion platforms, and attended MDT meetings to support participants in case of technical issues. The MDT telemeetings were held initially with Lifesize^TM^ software (Austin, TX, USA) and later with Microsoft Teams [37]. The standard requirements for using both platforms included a computer equipped with a digital camera and a microphone. No further devices, such as digitizers for microscopes, were required to conduct MDT telemeetings. The platforms allowed the creation of a virtual agenda to maintain the schedule of MDT meetings. On the day of appointment, MDT members accessed a virtual space that hosted the meeting, with the ability to share their screen and access the patient’s electronic medical record to consult the clinical documentation and discuss all the clinical findings relevant to the recommendation. This setting ensured the continuity of CCPs, even during the pandemic, along with the conduction of the MDT meetings, which shifted from face-to-face to teleconsultation format. By teleconsultation, we specifically mean a telematic approach defined as a medical act in which MDT participants converse remotely with each other via video call regarding a patient’s clinical situation, sharing clinical data, reports, images and audio-video materials related to the specific case [30]. 

The aim of this study was to examine whether the pandemic-induced implementation of teleconsultation for CCP MDT meetings supported overall MTD meeting performance and, in turn, the continuity of cancer care delivery by exploring how four MDT meeting indicators (i.e., (i) the number of members’ attendance, (ii) the number of cases to discuss, (iii) the duration of meetings, and (iv) the frequency of meetings) performed between 2019 and 2022.

## 2. Materials and Methods

This was a retrospective study on ten CCPs managed by the Quality and Accreditation Office, AUSL—IRCCS in Reggio Emilia, Italy. Ethical approval by the Institutional Review Board (IRB)/Ethics Committee was not applicable. CPs considered in the study involved neuro-oncology, pancreatic cancer, lung cancer, colorectal cancer, liver cancer, thyroid cancer, breast cancer, skin cancer, ovarian cancer, and lymphoma. 

To evaluate the performance of CCP MDT meetings in teleformat, we determined the average number of participants per year and the average number of cases discussed per year in each CCP as first and second endpoints of the study, respectively. The first endpoint was defined as the ratio between the total number of MDT meeting participants and the total number of meetings held, per year. The second endpoint was defined as the ratio between the total number of cases under discussion and the total number of meetings held, per year. We also determined the duration and frequency of MDT meetings as the third and fourth endpoints of the study, respectively. The third endpoint was defined as the average time between the discussion of the first and last case in each CCP MDT meeting, except for skin CCP due to reasons outside the purpose of the study. The fourth endpoint was defined as the difference between the a priori defined number of MDT meetings in each CCP and the number of meetings actually conducted over a one-year period. 

Over the study period, MDT meetings were held once a week with regard to the colorectal CCP, lymphoma CCP, breast CCP, ovarian CCP, pancreatic CCP, lung CCP, and skin CCP. In the neuro-oncology CCP, MDT meetings were held twice a month until 2021, and once a week starting from 2022. Liver CCP and thyroid CCP MDT meetings were held twice a month. 

Data on all four endpoints were obtained from electronic medical records produced at the end of each MDT meeting in 2019, 2020, 2021 and 2022, in which, according to standard practice, the meeting recommendations and outcomes, including frequency, participant attendance, cases to discuss, and length of discussion, are registered. These records are kept in an institutional repository and can be accessed by CCP data managers for indicator processing, which is returned annually to CCP members. It was not necessary to use different systems than those used before the teleformat was implemented to record all of the above information, including the MDT recommendations, in the electronic medical records. 

The discussions could involve incident cases, cases of recurrence, and/or individuals with a diagnostic suspicion of cancer. Each CCP establishes a priori criteria for selecting patients for discussion and for defining the appropriate phase of the pathway at which a patient should be discussed, according to regional regulations on cancer care. 

The endpoints related to 2019, i.e., before the pandemic, when MDT meetings were held face-to-face, were compared with the endpoints from 2020, i.e., during the major pandemic waves, when MDT meetings switched to virtual format, as well as from 2021 and 2022, i.e., after the major pandemic waves, when MDT meetings continued to be held in a virtual format or hybrid equivalent, to assess the support of teleconsultation in relation to MDT performance. Of note, since March 2020, teledevices have been promptly available for use. 

Statistical analysis was performed using R 4.2.2, R Core Team [38]. For each CCP, we summarized the mean session count of discussed cases and participants, accompanied by a Poisson 95% confidence interval, by year. We then explored the effect of year on mean count for each CCP, comparing the Poisson GLM model with and without year covariate by a likelihood ratio test. In case of a significant result (*p* < 0.05), we estimated the three-rate ratio (between the considered year in turn and 2019, the pre-pandemic year), accompanied by three 1–0.05/3 level two-tailed confidence intervals. We added a Bonferroni adjusted *p*-value against the null hypothesis of RR = 1 (no difference compared to 2019). For typographical reasons, only the latter is presented.

The third and fourth endpoints were analyzed by descriptive statistics.

## 3. Results

For each CCP, we first considered the overall test, which is statistically significant (*p* < 0.05) if there is at least one year (2019, 2020, or the following years) that is different from the others (*p* overall column in Table 1 and Table 2). Subsequently, only for those CCPs for which the overall test showed a significant difference, a comparison was made for each post-pandemic year (i.e., 2020, 2021, and 2022) to detect any change compared to the pre-pandemic year considered as baseline (i.e., 2019) (see *p* adjusted vs. 2019 column in Table 1 and Table 2).

### 3.1. First Endpoint

Table 1 shows the mean number of participants per year calculated for each CCP, together with a bilateral 95% confidence interval. We found a significant increase for colorectal CCP (*p* < 0.001; 2021 vs. 2019, *p* < 0.001; 2022 vs. 2019, *p* < 0.05), liver CCP (overall *p* < 0.05; 2021 vs. 2019, *p* < 0.001), ovary CCP (overall *p* < 0.001; 2021 vs. 2019, *p* < 0.001; 2022 vs. 2019, *p* < 0.001) and thyroid CCP (overall *p* < 0.001; 2021 vs. 2019, *p* < 0.001; 2022 vs. 2019, *p* < 0.001). A statistically significant decrease was found with regard to skin cancer CCP (overall *p* < 0.001; 2020 vs. 2019, *p* < 0.001), but in the remaining study period the mean estimate returned approximately to baseline. No significant differences were found for neuro-oncology, lymphoma, breast, pancreatic and lung CCPs.

### 3.2. Second Endpoint

Table 2 shows the mean number of cases discussed per year calculated for each CCP, together with a bilateral 95% confidence interval. We found a significant increase for neuro-oncology CCP (overall *p* < 0.001; 2021 vs. 2019, *p* < 0.001; 2022 vs. 2019, *p* < 0.05), ovary CCP (overall *p* < 0.001; 2020 vs. 2019, *p* < 0.001; 2021 vs. 2019, *p* < 0.001; 2022 vs. 2019, *p* < 0.001) and thyroid CCP (overall *p* < 0.001; 2022 vs. 2019, *p* < 0.001). A significant decrease was found with regard to liver CCP (overall *p* < 0.001; 2021 vs. 2019, *p* < 0.001; 2022 vs. 2019, *p* < 0.001) and skin cancer CCP (overall *p* < 0.001; 2020 vs. 2019, *p* < 0.001; 2021 vs. 2019, *p* < 0.05). No significant differences were found for colorectal, lymphoma, breast, pancreatic, and lung CCPs.

### 3.3. Third Endpoint

The duration of the CCP MDT meetings depend on the number of cases to discuss in addition to the complexity of each case. We calculated the average duration of MDT meetings in each CCP (see Table 3). The difference between meeting duration in 2019 and the longest/shortest meeting in the period 2020–2022 ranged from −00.39:00 to +00.31:00 min, with those meetings being related to lung and thyroid CPs, respectively (see Δ column in Table 3).

### 3.4. Fourth Endpoint

CCP data managers record requests from CCP MDT members regarding any changes in the frequency of MDT meetings with respect to what was established a priori. During the pandemic period, there were no requests for frequency-related changes except for the neuro-oncology and liver CCPs, which increased the number of monthly meetings. Based on the number of annual weeks, vacations, and holidays, the frequency of MDT meetings did not vary from the annual plan in any of the CCPs under study, considering an annual variation of ±6 meetings to be acceptable (see columns no. MDT meetings and frequency in Table 3).

## 4. Discussion

The first and second endpoints of this study suggest that the MDT meeting virtual format strengthened the overall CCP performance, accounting for an overall improved or not statistically significant different performance in nine out of ten (90%) CCPs for attendance and eight out of ten (80%) CCPs for number of cases discussed (Table 4a). Interestingly, although the teleformat implementation involved all CCPs, we observed different effects. In the case of the thyroid CCP and ovarian CCP, both endpoints improved, while in the case of the skin CCP, both endpoints worsened. During the pandemic, the skin CCP experienced a staff reduction in the Dermatology Department and a decrease in the number of patient admissions for suspected skin cancer, which probably determined the observed endpoints. In the case of the liver CCP, one endpoint worsened, while the other improved. In the case of the colorectal CCP and neuro-oncology CCP, one endpoint improved, while the other did not vary. In the case of the lymphoma CCP, breast CCP, pancreatic CCP and lung CCP, there was no effect in either endpoint (Table 4a).

Viewing the comparisons between 2019 and the following years (Table 4b), we see that the MDT meeting teleformat impacted the first endpoint to a greater extent than the second endpoint. In fact, we found that the percentage of positive significant changes concerning the first endpoint (7/8, 87.5%) was higher compared to that of the second endpoint (6/10, 60%). The overall distribution of positive significant changes increased from 2019 to 2022, and consequently, so did the number of CCPs involved, rising from one in 2020 to three in 2022. Over the study period, MDT meeting participants likely gained more experience in using the platform, overcame barriers to face-to-face attendance, and directly valued the benefits of telematic technology. In addition, we believe that it is important to consider non-significant changes as positive outcomes in light of the weight of the pandemic. 

Even for the liver CCP, the worsened endpoint did not appear to depend on the use of teleconferencing. In fact, the decrease in the average number of cases discussed is probably related to the increase in the number of MDT meetings, which has changed frequency from monthly to bimonthly since the last quarter of 2020.

With respect to the third endpoint, the observed changes in meeting duration did not appear to be related to the use of teleconferencing. Further investigation would be needed to verify the causes of the reported variations, which nevertheless remain acceptable.

Regarding the fourth endpoint, we would have expected a reduction in the frequency of meetings during the pandemic period. This did not occur, and leads us to hypothesize that CCP MDT members considered MDT meetings essential, and by extension, the telematic support to maintain them. 

Of note, the MDT meeting teleformat likely played an important role in maintaining the schedule of the lung CCP MDT meetings, despite the prominent involvement of Pneumology Department staff in the management of COVID-19 patients.

To date, the COVID-19 pandemic is seen as the impetus that gave dignity to telematic approaches, here intended as any virtual communication and/or conferencing between the parties involved in health care, which had previously been defined as a niche area unlikely able to replace the more traditional face-to-face MDT meeting format for cancer management [39,40]. To our knowledge, this is the first study to assess whether the teleformat for MDT meetings supported MDT performance through a comprehensive evaluation of more than one CCP. Over a longer study period, we observed results consistent with the study by Davis et al. [41] that reported higher attendance rates and a greater number of case presentations at virtual cancer MDT meetings [41]. Both before, and to a greater extent, after the pandemic, other studies have published results aligned with ours, but all of them refer to a smaller number of cases undergoing discussion, shorter study periods such as 6 months [42], individual disciplines, and/or CCP. Before the pandemic, Stevens et al. [43] demonstrated over a 6-month period that lung CCP cases discussed with the use of teleformats were not disadvantaged with respect to the recommended therapy or the appropriateness of decisions, compared to those discussed in the face-to-face format [43]. A study reporting on a 10-year experience in pediatric neuro-oncology videoconferencing provided evidence that the virtual approach was feasible and sustainable, leading to improvement in patients’ care with regard to the continuous effort to implement recommendations [44]. Van Huizen et al. [45] reported a mixed-method study over a 6-month period regarding a head and neck CCP, assessing that the added value of the videoconferenced MDT is small in terms of patient care, but MDT participants acknowledged that it is important to keep their medical viewpoints aligned and that their patients benefit from the discussions of complex cases [45]. Accordingly, by analyzing the survival rate following MDT discussion on patients affected by peritoneal mesothelioma, a national experience of monthly MDT videoconference meetings documented that the teleformat was effective at selecting patients suitable for specific treatments, favoring good outcomes from patient selection [46]. 

After the first pandemic waves, qualitative research better characterized the results of pre-pandemic studies, but still with reference to small study periods. Rajasekaran et al. [33] referred to sarcoma care, stating that the forced switch to virtual MDT meetings was an effective alternative to conventional face-to-face MDT meetings. The authors also underlined that the MDT meeting teleformat would facilitate conducting MDTs expanded to specialists abroad to seek opinions on complex cases, thus expanding oncology care globally [33]. Mohamedbhai et al. [47] demonstrated that head and neck CCP MDT participants felt able to perform in most indicators (such as data protection, decision making, technology, and organization/coordination), despite some concerns about the perception of a reduction in teamwork and training, as well as communication problems [47]. In 2022, Bonanno et al. [48] shared the results of a survey conducted among members of the European Society of Oncologic Imaging (ESOI) that explored the structure and efficacy of online MDT meetings during the COVID-19 pandemic, including benefits and limitations [48]. The findings confirmed that online MDT meetings are “a viable alternative to in-person meetings enabling continued timely high-quality provision of care with maintained coordination between specialties” [48]. By the time of writing, in a recent national mixed-method prospective cross-sectional study, the qualitative results showed that hybrid working and the possibility of virtual attendance through pandemic-related changes were positively maintained, against a series of limitations and objectives for future improvements [49]. The quantitative results showed significant improvement for MDT meeting organization and logistics compared to the access, case discussions, and patient representation [49].

Based on evidence coming from the pre and post-pandemic era on cancer care benefits related to the implementation of videoconferenced MDT meetings in CCPs, one might assume this implementation has optimized the overall quality of teamwork practice, mainly in terms of CCP MDT meeting attendance. An assessment of the underlying causes of pre-pandemic poor attendance at MDT meetings was not conducted, as it exceeds the scope of this study. MDTs working in cancer care require a substantial amount of the professionals’ time, and we believe that, in our setting, telediscussion has reduced the time needed for travel from provincial hospitals to the MDT meeting site, which favors member attendance. 

However, we are aware that this study has some limitations. First, the quality of the MDT meeting teleformat was not investigated, in terms of active rather than passive attendee contributions. Second, it was not possible to establish the weight of factors playing a role in defining intra-CCP differences, such as clinical characteristics of individual cancer types and/or attendees’ attitudes in preparing MDT meeting workflows [50], or inter-CCP differences, such as treatment complexity and/or the amount of incident and prevalent cases, which potentially affect the overall efficiency of a CCP. Furthermore, in two cases, namely the first endpoint of the lymphoma CCP and the second endpoint of the colorectal and pancreatic CCPs, we did not find significant differences from comparisons with 2019, despite the presence of a significant overall test. This, apart from a lower power of the statistical analysis for numerical reasons, may be because the observed overall difference came from comparisons between other years rather than with 2019. This critical issue could be caused by small differences between the total number of MDT meetings actually held and the expected number. These differences can be explained if we consider that some MDT meetings were postponed or rescheduled with additional sessions due to unforeseeable issues of daily clinical practice, such as the availability and amount of staff at that time in addition to the complexity of cases assigned for discussion.

The study limitations are counterbalanced by two-fold important strengths. First, at our center, CCPs and MDT meeting policy are regulated by a consolidated set of methodological rules for team functioning. Professional data managers ensure compliance with these rules, and, during the teleformat implementation period, also monitored and supervised CCP MDT members to make tele-implementation possible. Second, we confirmed that the MDT meeting teleformat positively influenced the CP model by maintaining effectiveness and increasing efficiency of CCP MDT meetings, while complying with pandemic-related restrictions. 

## 5. Conclusions

In conclusion, this study explored one aspect of teleconferencing use in health care in response to the COVID-19 pandemic, particularly in the field of cancer care and through the specific lens of MDT performance. Consistent with the assumption that it is necessary to understand which strategies adopted to face the COVID-19 crisis can still be useful after the pandemic and in what ways [51,52,53], we believe that the CCP MDT meeting teleformat represents an example of a pandemic-induced adaptation to take into account when planning models of care in the future. Accordingly, focusing on 2022, CCP MDT members wished to maintain the option to attend MTD meetings remotely, except for neuro-oncology CCP MDT members, who preferred to return to the in-person format. We are confident that the description of the consequences of teleconsulting implementation is just one of the different factors that can contribute to CCP functioning as a complex intervention. However, as the CP model aims to continuously improve quality care, insights from this study add another step toward this goal. Further studies are awaited to better characterize the advantages and disadvantages of implementing telematics in health care.

## Figures and Tables

**Table 1 cancers-15-02486-t001:** MDT performance with regard to the participants endpoint.

Care Pathway	Year	No. MDT Meetings	Total No. of Annual Attendance	Mean	Lower CI	Upper CI	*p* Overall	*p* Adjusted vs. 2019
Colorectal cancer	2019	47	558	11.9	10.9	12.9	<0.001 *	
	2020	53	620	11.7	10.8	12.6		1
	2021	49	747	15.2	14.2	16.4		<0.001 *
	2022	51	732	14.4	13.3	15.4		0.002 *
Neuro-oncology	2019	29	300	10.3	9.2	11.6	0.094	
	2020	27	245	9.1	8.0	10.3		
	2021	26	281	10.8	9.6	12.1		
	2022	39	428	11.0	10.0	12.0		
Liver cancer	2019	16	99	6.2	5.0	7.5	0.001 *	
	2020	19	155	8.2	6.9	9.5		0.095
	2021	23	226	9.8	8.6	11.2		<0.001 *
	2022	22	179	8.1	7.0	9.4		0.086
Lymphoma	2019	50	599	12.0	11.0	13.0	0.035	
	2020	48	541	11.3	10.3	12.2		0.911
	2021	50	653	13.1	12.1	14.1		0.381
	2022	47	609	13.0	12.0	14.0		0.519
Breast cancer	2019	51	773	15.2	14.1	16.3	0.203	
	2020	51	778	15.3	14.2	16.4		
	2021	52	844	16.2	15.2	17.4		
	2022	52	859	16.5	15.4	17.6		
Ovarian cancer	2019	48	243	5.1	4.5	5.7	<0.001 *	
	2020	47	279	5.9	5.3	6.7		0.209
	2021	51	363	7.1	6.4	7.9		<0.001 *
	2022	50	368	7.4	6.6	8.1		<0.001 *
Pancreatic cancer	2019	46	388	8.4	7.6	9.3	0.067	
	2020	49	416	8.5	7.7	9.3		
	2021	51	455	8.9	8.1	9.8		
	2022	45	333	7.4	6.6	8.2		
Lung cancer	2019	50	537	10.7	9.9	11.7	0.286	
	2020	51	540	10.6	9.7	11.5		
	2021	52	547	10.5	9.7	11.4		
	2022	52	605	11.6	10.7	12.6		
Skin cancer	2019	37	403	10.9	9.9	12.0	<0.001 *	
	2020	49	359	7.3	6.6	8.1		<0.001 *
	2021	48	449	9.4	8.5	10.2		0.080
	2022	48	489	10.2	9.3	11.1		0.961
Thyroid cancer	2019	22	126	5.7	4.8	6.8	<0.001 *	
	2020	19	136	7.2	6.0	8.4		0.214
	2021	25	294	11.8	10.5	13.2		<0.001 *
	2022	24	273	11.4	10.1	12.8		<0.001 *

*, statistical significance (*p* < 0.05); CI, confidence interval; MDT, multidisciplinary team; No., number.

**Table 2 cancers-15-02486-t002:** MDT performance with regard to the endpoint about cases under discussion.

Care Pathway	Year	No. MDT Meetings	Total No. of Discussed Cases	Mean	Lower CI	Upper CI	*p* Overall	*p* Adjusted vs. 2019
Colorectal cancer	2019	47	414	8.8	8.0	9.7	0.004	
	2020	53	426	8.0	7.3	8.8		0.554
	2021	49	497	10.1	9.3	11.1		0.102
	2022	51	435	8.5	7.8	9.4		1
Neuro-oncology	2019	29	256	8.8	7.8	10.0	<0.001 *	
	2020	27	285	10.6	9.4	11.8		0.114
	2021	26	337	13.0	11.6	14.4		<0.001 *
	2022	39	448	11.5	10.5	12.6		0.002 *
Liver cancer	2019	16	193	12.1	10.4	13.8	<0.001 *	
	2020	19	188	9.9	8.5	11.4		0.160
	2021	23	160	7.0	5.9	8.1		<0.001 *
	2022	22	164	7.5	6.4	8.7		<0.001 *
Lymphoma	2019	50	455	9.1	8.3	10.0	0.113	
	2020	48	412	8.6	7.8	9.4		
	2021	50	468	9.4	8.5	10.2		
	2022	47	474	10.1	9.2	11.0		
Breast cancer	2019	51	977	19.2	18.0	20.4	0.311	
	2020	51	964	18.9	17.7	20.1		
	2021	52	1041	20.0	18.8	21.3		
	2022	52	1057	20.3	19.1	21.6		
Ovarian cancer	2019	48	126	2.6	2.2	3.1	<0.001 *	
	2020	47	195	4.1	3.6	4.8		<0.001 *
	2021	51	246	4.8	4.2	5.5		<0.001 *
	2022	50	283	5.7	5.0	6.3		<0.001 *
Pancreatic cancer	2019	46	216	4.7	4.1	5.3	<0.001	
	2020	49	191	3.9	3.4	4.5		0.183
	2021	51	284	5.6	4.9	6.2		0.177
	2022	45	248	5.5	4.9	6.2		0.256
Lung cancer	2019	50	694	13.9	12.9	14.9	0.073	
	2020	51	628	12.3	11.4	13.3		
	2021	52	708	13.6	12.6	14.6		
	2022	52	728	14.0	13.0	15.0		
Skin cancer	2019	37	439	11.9	10.8	13.0	<0.001 *	
	2020	49	399	8.1	7.4	9.0		<0.001 *
	2021	48	451	9.4	8.6	10.3		0.002 *
	2022	48	543	11.3	10.4	12.3		1
Thyroid cancer	2019	22	113	5.1	4.2	6.1	<0.001 *	
	2020	19	126	6.6	5.5	7.9		0.146
	2021	25	148	5.9	5.0	6.9		0.767
	2022	24	235	9.8	8.6	11.1		<0.001 *

*, statistical significance (*p* < 0.05); CI, confidence interval; MDT, multidisciplinary team; No., number.

**Table 3 cancers-15-02486-t003:** MDT performance with regard to meeting duration express as hh: mm, and frequency.

Care Pathway	Year	No. MDT Meetings	Frequency	Mean Meeting Duration	Overall Mean	Δ
Colorectal cancer	2019	47	1/w	01:10	01:07	−00:08
	2020	53		01:10		
	2021	49		01:07		
	2022	51		01:02		
Neuro-oncology	2019	29	2/m	01:24	01:20	−00:25
	2020	27		01:29		
	2021	26		01:31		
	2022	39	1/w	00:59		
Liver cancer	2019	16	1/m	01:28	01:09	−00:34
	2020	19		01:03		
	2021	23	2/m ^§^	01:12		
	2022	22		00:54		
Lymphoma	2019	50	1/w	01:15	01:14	−00:04
	2020	48		01:14		
	2021	50		01:11		
	2022	47		01:16		
Breast cancer	2019	51	1/w	02:01	02:16	+00:30
	2020	51		02:11		
	2021	52		02:31		
	2022	52		02:21		
Ovarian cancer	2019	48	1/w	00:38	00:44	+00:11
	2020	47		00:49		
	2021	51		00:41		
	2022	50		00:48		
Pancreatic cancer	2019	46	1/w	01:21	01:09	−00:21
	2020	49		01:16		
	2021	51		01:02		
	2022	45		01:00		
Lung cancer	2019	50	1/w	02:11	01:49	−00:39
	2020	51		01:53		
	2021	52		01:32		
	2022	52		01:41		
Skin cancer	2019	37	1/w	Na	Na	Na
	2020	49				
	2021	48				
	2022	48				
Thyroid cancer	2019	22	2/m	01:09	01:17	+00:31
	2020	19		01:12		
	2021	25		01:09		
	2022	24		01:40		

^§^, since the last trimester of 2020; m, month; Na, not available; w, week; Δ, difference between meeting duration in 2019 and the longest/shortest meeting in the period 2020–2022.

**Table 4 cancers-15-02486-t004:** Schematic representation of MDT performance with regard to overall significance (Table 4a) and significance between 2019 and 2020, 2021 and/or 2022 (Table 4b) of study endpoints.

(a) Overall Significance of Study Endpoints						
Cancer Care Pathway	No. of Participants			No. of Cases		
Thyroid cancer	↑			↑		
Ovarian cancer	↑			↑		
Skin cancer	↓			↓		
Liver cancer	↑			↓		
Colorectal cancer	↑			n.s.		
Neuro-oncology	n.s.			↑		
Lymphoma	n.s.			n.s.		
Breast cancer	n.s.			n.s.		
Pancreatic cancer	n.s.			n.s.		
Lung cancer	n.s.			n.s.		
**(b) Significance between 2019 and 2020, 2021 and/or 2022 (Table 4b) of Study Endpoints**						
**Cancer Care Pathway**	**No. of Participants**			**No. of Cases**		
	**2020 vs. 2019**	**2021 vs. 2019**	**2022 vs. 2019**	**2020 vs. 2019**	**2021 vs. 2019**	**2022 vs. 2019**
Thyroid cancer	n.s.	↑	↑	n.s.	n.s.	↑
Ovarian cancer	n.s.	↑	↑	↑	↑	↑
Skin cancer	↓	n.s.	n.s.	↓	↓	n.s.
Liver cancer	n.s.	↑	n.s.	n.s.	↓	↓
Colorectal cancer	n.s.	↑	↑	n.s.	n.s.	n.s.
Lymphoma	n.s.	n.s.	n.s.	n.s.	n.s.	n.s.
Breast cancer	n.s.	n.s.	n.s.	n.s.	n.s.	n.s.
Neuro-oncology	n.s.	n.s.	n.s.	n.s.	↑	↑
Pancreatic cancer	n.s.	n.s.	n.s.	n.s.	n.s.	n.s.
Lung cancer	n.s.	n.s.	n.s.	n.s.	n.s.	n.s.

↑, Positive change; ↓, Negative change; n.s., not statistically significant change.

## Data Availability

Data are available upon reasonable request to the corresponding author due to privacy restrictions.

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
