# Peer review of "Learning from Adaptations to the COVID-19 Pandemic: How Teleconsultation Supported Cancer Care Pathways at a Comprehensive Cancer Center in Northern Italy"

_cancers, 2023, doi:10.3390/cancers15092486_

Round 1
Reviewer 1 Report
Dear authors,
I would like to extend my congratulations to you on the publication of your paper "Learning from Adaptations to the COVID-19 Pandemic: How 2 Teleconsultation Supported Cancer Care Pathways at a Comprehensive Cancer Center in Northern Italy". Your study on the implementation of teleconsultation in multidisciplinary team (MDT) meetings related to cancer care pathways (CCPs) during the COVID-19 pandemic is a timely and important contribution to the field of healthcare.
The results of your study are particularly noteworthy, as they demonstrate that teleconsultation supported the CCPs and the delivery of cancer care during the pandemic. I believe that your findings will have a major impact on the use of telematic tools in the future, not only in the context of cancer care but also in other areas of healthcare.
Once again, congratulations on your excellent work, and thank you for your valuable contribution to the field of healthcare.
Best regards,
Author Response
We are grateful to the Reviewer for dedicating his/her time to read our article. We thank you for your comments.
Reviewer 2 Report
A well written manuscript that would be of interest to other cancer services which faced the same issues during the COVID 19 pandemic.
1. Some suggestions that would add to the practical benefit of this article.
a). Some description of the technical issues that had to be overcome in the initial phases of roll out of the teleconsultation in the MDT meetings. For example what videoconferencing platform was used and was there a need to obtain cameras or digitizers for microscope to show histopathology. Were different systems required to record recommendations of the MDTs?
b) Sharing the set of methodological rules for MDT Team functioning used by the Local Health Authority in a Supplementary materials section would be useful for readers.
2. A comment as to whether cancer services were compromised in the region during the initial phases of the pandemic in 2020 and also during subsequent waves that could have affected case numbers. In some countries and jurisdictions there were quite significant compromise in suspension of screening programs, surgical lists, access to diagnostic services including endoscopy and radiology leading to delays in diagnosis and treatments with a subsequent catch up.
3.Reference 30 URL is broken and does not resolve.
4. Line 323: I think the authors mean the time needed for travel from provincial hospitals.
English is of excellent standard.
Author Response
A well written manuscript that would be of interest to other cancer services which faced the same issues during the COVID 19 pandemic.
We thank the Reviewer for dedicating his/her time to critically read and comment our manuscript. Your comments were precious to improve it.
Point 1: Some suggestions that would add to the practical benefit of this article.
a). Some description of the technical issues that had to be overcome in the initial phases of roll out of the teleconsultation in the MDT meetings. For example what videoconferencing platform was used and was there a need to obtain cameras or digitizers for microscope to show histopathology. Were different systems required to record recommendations of the MDTs?
Response 1a: Thank you for this comment. Accordingly, we have now amended the text from line 127 to line 135, stating that "The MDT telemeetings were held initially with LifesizeTM software and later with Microsoft Teams [37]. The standard requirements for using both platforms included a computer equipped with a digital camera and a microphone. No further devices, such as digitizers for microscope, were required to conduct MDT telemeetings. The platforms allowed the creation of a virtual agenda to maintain the schedule of MDT meetings. On the day of appointment, MDT members accessed a virtual space that hosted the meeting, with the ability to share the screen and access patient's electronic medical record to consult the clinical documentation and discuss all the clinical findings relevant to the recommendation.". The reference list has been modified accordinging the added reference.
We have also amended the text from line 174 to line 180, stating that MDT recommendations were registered in electronic medical records by the same system used before the implementation of teleformat.
b) Sharing the set of methodological rules for MDT Team functioning used by the Local Health Authority in a Supplementary materials section would be useful for readers.
Response 1b: Thank you for this suggestion. We have now produced a Supplementary file, called Supplementary file 1, which describes the methodological rules for MDT functioning used by our Institution.
In the revised version of the manuscript, the Supplementary file 1 was cited in lines 112-113.
Point 2: A comment as to whether cancer services were compromised in the region during the initial phases of the pandemic in 2020 and also during subsequent waves that could have affected case numbers. In some countries and jurisdictions there were quite significant compromise in suspension of screening programs, surgical lists, access to diagnostic services including endoscopy and radiology leading to delays in diagnosis and treatments with a subsequent catch up.
Response 2: Thank you for this suggestion. Accordingly, we have now amended the text from line 96 to line 109, stating that "Both the province and region where our reality is located were hit severely and early by the first wave of the pandemic. To face the challenges the pandemic posed, our health system implemented a specific strategy based on the following key elements: building separate pathways for COVID and non-COVID patients; treating all COVID patients in need of hospitalization, while continuing to ensure adequate and appropriate care to all non-COVID patients in case of emergency and/or complex cases; never stop delivering of cancer care; and moving promptly and in a coordinated way within the network and in cooperation with other local health organizations. As reported by researchers of our center [36], screening programs were suspended from the middle of March to the end of May on average but, once resumed, actions implemented to screen patients scheduled in that period allowed the impact of the lockdown on cancer screening delays, and hence diagnosis, to be minimized. New diagnoses returned to a number only slightly lower than those observed in 2019.". The reference list has been modified accordinging the added reference.
Point 3: Reference 30 URL is broken and does not resolve.
Response 3: Thank you for this observation. Reference 30 URL contained a hyphenation error that was corrected in the revised version of the manuscript.
Point 4: Line 323: I think the authors mean the time needed for travel from provincial hospitals.
Response 4: Thank you for this suggestion. We have now replaced the term "transfer" with the term "travel".